# Ecological Implications in a Human-Impacted Lake—A Case Study of Cyanobacterial Blooms in a Recreationally Used Water Body

**DOI:** 10.3390/ijerph20065063

**Published:** 2023-03-13

**Authors:** Agnieszka Napiórkowska-Krzebietke, Julita Anna Dunalska, Elżbieta Bogacka-Kapusta

**Affiliations:** 1Department of Ichthyology, Hydrobiology and Aquatic Ecology, National Inland Fisheries Research Institute, Oczapowskiego 10, 10-719 Olsztyn, Poland; 2Institute of Geography, Faculty of Oceanography and Geography, University of Gdańsk, Jana Bażyńskiego 8, 80-309 Gdańsk, Poland; 3Department of Lake Fisheries, National Inland Fisheries Research Institute, Rajska 2, 11-500 Giżycko, Poland

**Keywords:** cyanobacterial bloom, cyanotoxin, *Sphaerospermopsis aphanizomenoides*, *Cuspidothrix issatschenkoi*, *Raphidiopsis raciborskii*

## Abstract

This study was aimed primarily at describing the planktonic assemblages with special attention to invasive and toxin-producing cyanobacterial species in the context of ecological and health threats. The second aim was to analyze the aspect of recreational pressure, which may enhance the cyanobacterial blooms, and, as a consequence, the negative changes and loss of planktonic biodiversity. This study was carried out in recreationally used Lake Sztynorckie throughout the whole growing season of 2020 and included an assessment of the abundance and biomass of phytoplankton (cyanobacteria and algae) in relation to environmental variables. The total biomass was in the range of 28–70 mg L^−1^, which is typical for strong blooms. The dominant filamentous cyanobacteria were *Pseudanabaena limnetica*, *Limnothrix redekei*, *Planktolyngbya limnetica*, and *Planktothrix agarhii*, and three invasive nostocalean species *Sphaerospermopsis aphanizomenoides*, *Cuspidothrix issatschenkoi*, and *Raphidiopsis raciborskii*. They can pose a serious threat not only to the ecosystem but also to humans because of the possibility of cyanobacteria producing cyanotoxins, such as microcystins, saxitoxins, anatoxin-a, and cylindrospermopsins, having hepatotoxic, cytotoxic, neurotoxic, and dermatoxic effects. The water quality was assessed as water bodies had bad ecological status (based on phytoplankton), were highly meso-eutrophic (based on zooplankton), and had very low trophic efficiency and low biodiversity.

## 1. Introduction

Recently, the harmful cyanobacterial blooms, also called “cyanoHABs”, have been recognized as increasing worldwide due to nutrient enrichment connected with progressive eutrophication, global climate warming, and even hydrological alteration. The cyanobacteria, including toxin-producing and bloom-forming species, are of special concern because of their harmfulness. The most frequent toxin-producing cyanobacterial blooms were connected with the occurrence of both nitrogen-fixing and non-nitrogen-fixing species of the genera *Aphanizomenon* and *Dolichospermum* or *Microcystis* and *Planktothrix* [1,2,3,4]. The analysis of cyanotoxins (CTs) is primarily focused on microcystins, which are of special concern because of their effect on human health in a guideline value of 1 μg L^−1^ for MC-LR in drinking water [5]. In addition to microcystins, other toxins posing a health risk, such as nodularins, cylindrospermopsin, anatoxins, and saxitoxins, are produced by many cyanobacterial species [6].

Cyanobacteria can cause local biodiversity loss, threatening the functioning of the whole ecosystem with serious ecosystem damage [7,8]. In the temperate zone, some of them are known worldwide as invasive, harmful, not-native, or even exotic species, especially *Sphaerospermopsis aphanizomenoides* (Forti) (Zapomelová, Jezberová, Hrouzek, Hisem, Reháková, and Komárková), *Cuspidothrix issatschenkoi* (Usachev) (P.Rajaniemi, Komárek, R.Willame, P. Hrouzek, K.Kastovská, L.Hoffmann, and K.Sivonen), and *Raphidiopsis raciborskii* (Wołoszyńska) (Aguilera, Berrendero Gómez, Kaštovský, Echenique, and Salerno [9,10,11,12]). All three of these species belong to Invasive Nostocalean Cyanobacteria, which can occur together [13]. In some countries, especially in Australia, New Zealand, Brazil, China, and Japan, these cyanobacteria species were reported as producers of cyanotoxins, such as saxitoxin, anatoxin-a, and cylindrospermopsin [11,14,15,16]. Recently, a toxic *C. issatchenkoi* has also been recorded in a river in Kazakhstan [17]. According to Ballot et al. [18], *C. issatschenkoi* is also responsible for producing anatoxin-a, a neurotoxic bicyclic secondary amine [11,19]. Cylindrospermopsin is well-known as the second most frequently listed cyanotoxin being cytotoxic, dermotoxic, and hepatotoxic cyclic guanidinic alkaloid [20], also produced by *R. raciborskii* and *C. issatschenkoi* [21]. Saxitoxin, in turn, is a neurotoxic guanidinium derivative with two amine functional groups. The findings of Sabour et al. [22] and Dziga et al. [23] confirm the ability of *S. aphanizomenoides* to produce three variants of microcystins.

Eutrophication and especially elevated phosphorus concentrations promote the occurrence and/or stimulate the expansion of invasive species, such as *S. aphanizomenoides*, which can grow in poor light conditions with high phytoplankton biomass [10]. The human-affected lakes are generally characterized by progressive eutrophication related to high phytoplankton biomass and a high share of cyanobacteria, e.g., urban lakes exposed to direct or indirect sources of pollution [24,25]. Recently, water-related recreation has also been treated as serious pollution, which aggravates the adverse changes in water quality. The negative effects of recreational activities on water quality were confirmed in a few findings [26,27]. The small, shallow, and recreationally used Lake Sztynorckie was selected to present the pattern of impact on the water environment. In 2020, the tourism pressure, the number of users (chartering and passengers) from the port was as follows: in April—583 people; in May—5015 people; June—14,724 people; in July—15,880 people; and in August—15,822 people (data from the port owner’s register). Tourist activities, including chartering the yachts with the function of transport and accommodation for passengers, occur directly to enhance the pollutants in the water [28].

Impacts on lakes from tourist activities occur directly to the lake water and shoreline or can affect the water body indirectly through various actions in the catchment [29]. The booming tourism activity is chartering yachts. They can fulfil the function of transport and accommodation for passengers and are, therefore, aimed at consumers with middle-income, young, and middle-aged tourists [30]. All infrastructure for yachts can have significant impacts on the sediment and tissue of organisms.

Therefore, the aims of this study were to (1) describe the phytoplankton and zooplankton assemblages with special attention to invasive toxin-producing cyanobacteria species and (2) to investigate if cyanobacterial blooms linked with loss of planktonic biodiversity and changes in the whole phytoplankton and zooplankton structures may be exacerbated by recreational pressure.

## 2. Materials and Methods

Lake Sztynorckie (54°7′42.187″ N 21°40′40.44″ E) is located in the Masurian Lakes District of north-eastern Poland (Figure 1). It is a small (surface area of 47.3 ha, volume of 906,900 m^3^), lowland, non-stratified, very shallow (the max depth is 3.1 m, mean depth is 1.9 m) lake. This lake is used by sailors, and it is connected with Lake Dargin by a short, 170-m-long Sztynorcki channel. The lake catchment belongs to the waters of the Mamry Lake complex discharged by River Węgorapa to River Pregoła. According to the Computer Map of the Hydrographic Division of Poland (The State Water Holding Polish Waters), Lake Sztynorckie has one main tributary, which is divided into three sections: (1) a permanent inflow (a drainage ditch from the catchment area); (2) flow through Lake Sztynorckie; and (3) a channel connecting Lake Sztynorckie with Lake Dargin.

The water samples for standard analysis used in a system for routine monitoring of water quality were taken from a representative site S1 at the deepest point of the lake. Additionally, the sub-surface water samples were taken from five sites, 1–5, i.e., site S1—profundal zone as the deepest point of the lake, site S2—inflow zone near channel between the pumping station and Lake Sztynorckie (outflow from the retention reservoir), site S3—channel zone including Sztynorcki channel, which connects Lake Sztynorckie with Lake Dargin, and sites associated with the harbour, i.e., site S4—fuel zone, and site S5—mooring zone. These five sampling sites were selected to check the impact of human activities on the waters.

The water samples for planktonic organisms, including phytoplankton (cyanobacteria and algae) and zooplankton (rotifers, copepods, cladocerans), were collected in April, June, August, and October 2020, according to methods used for routine monitoring of biological elements. These methods include the integrated water samples, which were taken at the deepest site S1 from the whole water column, i.e., at one-meter intervals, mixed in the bucket, and then the final water sample was taken. Additional samples were taken from the subsurface (0.5 m) water layers. The phytoplankton samples for biomass analysis were not concentrated, whereas zooplankton samples were concentrated (25 L), using a 55 µm mesh size plankton net. These samples were fixed with Lugol’s solution and 96% ethanol. Furthermore, for phytoplankton taxonomic analysis, the samples were taken using a 10 µm mesh size plankton net, and they were analyzed live after sampling.

The phytoplankton density and biomass were analyzed using an inverted microscope according to the Utermöhl method [31]. The biomass was calculated via the cell biovolume measurements according to standard methods revised in Napiórkowska–Krzebietke and Kobos [32]. The phytoplankton structure was determined using a light microscope and magnifications of 200×, 400×, and 1000× with oil immersion. Species identifications followed the latest references [33,34,35,36,37], and their currently accepted taxonomic names were confirmed according to Algaebase [38]. Special attention was paid to the presence of three invasive species, especially *Sphaerospermopsis aphanizomenoides*, *Cuspidothrix issatschenkoi*, and *Raphidiopsis raciborskii*. Published data focused on the presence of these species without indicating the possibility of them occurring simultaneously in the same environment.

The density of zooplankton was quantified using a Sedgwick–Rafter counting cell under an optical microscope. Identification of zooplankton species was based mainly on Flössner [39], Radwan et al. [40], and Rybak and Błędzki [41]. The biomass calculations were performed on length and length–dry mass relationships for rotifers, according to Ejsmont–Karabin [42], and for crustaceans, according to Bottrell et al. [43].

The data on physicochemical parameters, including water temperature, dissolved oxygen, electrolytic conductivity, pH, nitrites, nitrates, ammonium nitrogen, total nitrogen, total phosphorus, phosphates, total organic carbon, chlorophyll a, and total suspended solids, were collected on the same dates and from the same sites as for planktonic parameters [44]. These data were used to analyze the planktonic organisms’ responses to the environmental parameters and to test any similarities or differences between the examined objects using statistical tools.

The ecological status assessment of Lake Sztynorckie was based on calculations of the Phytoplankton Metric for Polish Lakes (PMPL), according to the equations adapted from Napiórkowska-Krzebietke et al. [45] and references therein. The water quality was classified in line with the Polish Regulations [46]. The trophic state of Lake Sztynorckie was assessed based on zooplankton, trophic state index based on Rotifera TSI_ROT_ [42], and based on Crustacea TSI_CRUS_ [47]. Furthermore, water quality was assessed based on biodiversity indices, i.e., species richness R, Shannon–Weaver diversity index S–WI [48], and evenness [49], which were calculated via the density of phytoplankton and zooplankton. Determination of the main phytoplankton representatives of functional groups was based on the functional classification confirming their specific habitat requirements [50,51].

Trophic efficiency (TE) was used to describe the relations between zooplankton and phytoplankton, and the total zooplankton biomass to total phytoplankton biomass ratio was calculated. The classification of the efficiency level followed the class thresholds given in Napiórkowska-Krzebietke [52], i.e., from the lowest to the highest—maximum of: 0.20 for class I, 0.40 for class II, 0.60 for class III, 0.80 for class IV, and 1.00 for class V.

The biological parameters were checked with the Shapiro–Wilk test for normality, and the data were not normally distributed. The nonparametric Kruskal–Wallis test was used to check the significant differences between parameters at five sites. The significance level was set to *p* < 0.05. The relationships between parameters were tested with Spearman’s rank correlation coefficient. All these analyses were performed with Statistica software (ver. 13.3 for Windows, Statsoft, Tulsa).

Redundancy analysis (RDA) was used to analyze the planktonic organisms’ response to environmental parameters. Therefore, a Monte Carlo test with 999 random permutations (*p* < 0.05) was applied to choose the most important variables. Next, the dissimilarity between sampling sites was tested with the non-metric multidimensional scaling (NMDS) ordination analysis with the Bray–Curtis distance measure. Stress formula type 2 was applied for the log-transformed physicochemical and biological parameters. All analyses were performed using Canoco for Windows 4.5 software.

## 3. Results

### 3.1. Phytoplankton and Zooplankton Assemblages within Routine Monitoring System

Phytoplankton assemblages were built by 56–76 taxa belonging to eight phyla, including Cyanobacteria, Bacillariophyta, Charophyta, Chlorophyta, Cryptophyta, Euglenozoa, Miozoa, and Ochrophyta. At the deepest representative site of the lake, the phytoplankton density ranged from 35.5 × 10^6^ to 86.3 × 10^6^ ind. L^−1^ throughout the growth season (Figure 2A), with the minimum in June and the maximum in August. The highest density was always recorded for Cyanobacteria, i.e., 72–89%, followed by the share of 6–20% for Chlorophyta and 15% in April for Bacillariophyta. The remaining taxonomic groups were much less abundant.

A similar tendency was noted concerning the total phytoplankton biomass, which ranged from approximately 32 mg L^−1^ (June) to 70 mg L^−1^ (August) (Figure 2B). Dominating Cyanobacteria formed biomass of about 24–57 mg L^−1^, i.e., 67–82%. The shares of other main taxonomic groups were as follows: 4–20% for Bacillariophyta, 1–16% for Cryptophyta, and 2–7% for Chlorophyta.

The ecological status assessment based on the phytoplankton index indicated the bad ecological status of Lake Sztynorckie (PMPL = 4.5).

Dominant cyanobacterial species *Limnothrix redekei* (Goor) (Meffert), *Pseudanabaena limnetica* (Lemmermann) (Komárek), and *Planktolyngbya limnetica* (Lemmermann) (Komárková-Legnerová and Cronberg) were present during the whole growing season. Their shares in total biomass were 7–48%, 10–28%, and 5–13%, respectively (Table 1). The less abundant were: *Aphanizomenon gracile* (Lemmermann), *Raphidiopsis raciborskii* (Woloszynska) (Aguilera, Berrendero Gómez, Kastovsky, Echenique and Salerno), and *Planktothrix agardhii* (Gomont) (Anagnostidis and Komárek).

In April, diatoms *Ulnaria acus* (Kützing) (Aboal) dominated with a share of 20% and *Ulnaria ulna* (Nitzsch) (Compère) with a share of 5%. In October, *Cryptomonas erosa* (Ehrenberg) formed 16% of the total biomass. The growth rate of other species was limited.

Zooplankton assemblages were formed by 37 species belonging to Rotifera, Cladocera, and Copepoda. The total zooplankton density ranged from 1177 ind. L^−1^ to 3768 ind. L^−1^ with the minimum in April and the maximum in August (Figure 3A). Rotifera was always the dominant group, with shares of 75–99% in the total density. In June, the shares of 15% and 10% belonged to Cladocera and Copepoda, respectively. The presence of Cladocera in zooplankton was not recorded in August and October.

Zooplankton formed total biomass in the range of 0.5–5.7 mg L^−1^ (Figure 3B). The biomass of Rotifera was relatively stable (0.3–0.6 mg L^−1^), but its percentage share in the total biomass changed from 6% in June to 96% in October. In June, the highest biomass was that of Cladocera (3.2 mg L^−1^; 56%), with dominant species of *Bosmina longispina* Leydig and *Bosmina longirostris* O.F. Müller (Table 2), followed by Copepoda, where the species *Macrocylops albidus* Jurine, and *Thermocyclops* sp. with the second group at relatively high biomass (2.1 mg L^−1^; 38%). Rotifera dominated in the remaining months, and the dominant species were as follows: *Polyarthra longiremis* Carlin; *Keratella quadrata* Müller; *Asplanchna priodonta* Gosse; and *Anuraeopsis fissa* Gosse.

The overall assessment according to zooplankton indicated a highly meso-eutrophic level based on Rotifera (TSI_ROT_ = 53.1), and mesotrophic level based on Crustacea (TSI_CRUS_ = 34.1).

### 3.2. Spatial Changes in Phytoplankton and Zooplankton Assemblages

The distribution of total phytoplankton density in surface waters indicated high variability in spatial and temporal scales. The highest average density (58.0 × 10^6^ ind. L^−1^) was recorded at sites S5 and S4, and the lowest (46.8 × 10^6^ ind. L^−1^) at site S3 (Figure 4A). However, nonsignificant (*p* > 0.05) changes were recorded in spatial distribution. Temporal changes showed that maximum density was noted in August at sites S1, S4, and S5 or in April (S2) and in October (S4). The lowest density was in June at all sites. The dominant group was always Cyanobacteria, with shares of 68–85% of the total density. Chlorophyta, with 5–27%, and Bacillariophyta, with 1–18%, were the most abundant algae groups.

Similar tendencies in spatial distribution, with maxima of 42.8 mg L^−1^, on average, at sites S4 and S5, and a minimum of 32.5 mg L^−1^, on average, at site S3, were noted in phytoplankton biomass (Figure 4B). On a temporal scale, the highest biomass of 62.6 and 61.4 mg L^−1^ were recorded in April and August at sites S2 and S1, respectively. The lowest biomass of 15.6 mg L^−1^ was in June at site S3. Based on the biomass, the dominant group was also Cyanobacteria (52–87%). Among algae, only Bacillariophyta and Cryptophyta had relatively high shares, i.e., in the range of 2–35% and 1–10%, respectively.

Zooplankton density changed from 5 to 38,709 ind. L^−1^ (3227 ind. L^−1^, on average) (Figure 5A) with a high coefficient of variability (CV > 200%). Spatially, the lowest average density was noted at site S3 (954 ind. L^−1^) and the highest at site S2 (10,111 ind. L^−1^). Regarding temporal changes, the maximum density was always noted in August, and the minimum in October, especially at site S5 (only 5 ind. L^−1^). The main numerical contributor to the zooplankton structure was Rotifera, except for site S3 in June, with the domination of cladoceran species.

The total zooplankton biomass changed from 0.01 to 2.40 mg L^−1^ (Figure 5B) with similar averages at sites S1–S4 (0.72–0.76 mg L^−1^) and half of that value at site S5. In spatial scale, the maximum was at site S4, and the minimum was at site S5. On the temporal scale, the highest biomass was noted in June at S4 due to the presence of Cladocera at sites S1, S3, and S5, and Cladocera and Copepoda at site S4. A notable exception: the highest total biomass and Rotifera biomass were found at site S2 in August. Concerning both the total zooplankton density and biomass, their values were statistically and spatially differentiated (KW-H(4;20) = 11.2429; *p* = 0.0240 and KW-H(4;20) = 10.3571; *p* = 0.0348 for density and biomass, respectively).

### 3.3. Ecological Implications: Phytoplankton–Zooplankton–Environment

To gain knowledge about the planktonic organisms’ response to environmental parameters, the following physicochemical parameters were of particular interest: water temperature, dissolved oxygen, conductivity, pH, nitrites, nitrates, ammonium nitrogen, total nitrogen, total phosphorus, phosphates, total organic carbon, chlorophyll a, and total suspended solids. The redundancy analysis (RDA) showed strong positive relations between the total phytoplankton biomass and biomass of selected phyla with pH, TSS, Chl-a, TN, DO, and TOC (Figure 6). Concerning zooplankton, the total biomass and biomass of Copepoda and Cladocera were positively correlated primarily with water temperature and TP. The relations between phytoplankton and zooplankton were negative.

The zooplankton-to-phytoplankton relations were also expressed as the biomass ratio, i.e., trophic efficiency. The TE values ranged from 0.01 to 0.18 and were characteristic of very low trophic efficiency at all sites.

The NMDS analysis was performed separately on the environmental parameters (Figure 7A) and planktonic parameters (Figure 7B) for all sampling sites and terms, and it provided two different sample arrangements. In the first assumption, the samples taken in April and October were primarily arranged by the majority of analyzed abiotic parameters, especially by Chl-a, pH, TN, and TOC (Figure 7A), while water temperature was the key factor for the distribution of summer samples. The samples with distinct separation concerned site S2 (inflow zone) in April and June related to Cond, TOC, and TP.

In the second assumption (Figure 7B), the phytoplankton biomass was opposed to zooplankton biomass. The samples taken in August and October were distributed along a path within the NMDS 1, the majority of phytoplankton groups, especially by Cyanobacteria, Cryptophyta and Chlorophyta, and zooplanktonic Rotifera. All of the spring samples (April) were primarily arranged by Bacillariophyta along NMDS2. The samples taken in June were separated and associated with zooplankton and especially with Cladocera and Copepoda. Similarly, for both assumptions, the samples from the inflow zone (S2) were usually distinctly separated from the others, whereas samples from channel and fuel zones (S3 and S4) tended to be closely gathered.

### 3.4. Potential Toxin-Producing Cyanobacteria, Including Invasive Species

In Lake Sztynorckie, the dominant toxin-producing cyanobacteria species were *L. redekei*, *P. limnetica*, *A. gracile*, and *P. limnetica*. Among them, three invasive species, *R. raciborskii*, *S. aphanizomenoides*, and *C. issatschenkoi*, frequently co-occurred. However, their occurrence and biomass differed in both spatial and temporal scales. In April and June, *S. aphanizomenoides* and *C. issatschenkoi* were present at S4 or S5, i.e., sites connected with fuel and mooring zones. In August, in turn, all these species co-occurred at all sampling sites (Table 3).

## 4. Discussion

The small and very shallow Lake Sztynorckie is a recreationally used lake located in the biggest Polish Lakeland. According to potential pollution sources, the lake was divided into five sampling zones, including profundal, inflow, channel fuel, and mooring zones. Generally, high total phytoplankton biomass ranged within 28–70 mg L^−1^, typical for strong blooms which lasted throughout the whole growing season, similar to other small water bodies under strong pressure [8,24,25,53,54]. Such high biomass is characteristic of lakes with progressive eutrophication. The main cyanobacterial bloom contributors in Lake Sztynorckie were filamentous cyanobacteria *Pseudanabaena limnetica*, *Limnothrix redekei*, *Planktolyngbya limnetica*, and *Aphanizomenon gracile*. They are the representatives of S1 and H1 coda, which are typical of turbid mixed environments, including shade-adapted cyanobacteria, and of eutrophic, stratified, and non-stratified lakes with low nitrogen content, respectively [50,51].

Generally, dominant and potentially toxic or toxic cyanobacterial species of the genera *Aphanizomenon*, *Raphidiopsis*, *Cuspidothrix*, *Dolichospermum*, *Sphaerospermopsis*, *Planktothrix*, *Limnothrix*, *Pseudanabaena*, and *Planktolyngbya* were also observed in Lake Sztynorckie [6]. Thus, there was the risk of producing toxins that have hepatotoxic, dermatoxic, neurotoxic, and cytotoxic effects on other organisms. Some specific reactions concern toxic effects on liver cells through dermal damage and inhibition of protein synthesis to damage the structure of DNA and RNA [55]. These cyanobacteria concern primarily people engaging in water recreation and cause negative effects on their health from recreational exposure [56]. Among these taxa, the three invasive nostocalean species, *Sphaerospermopsis aphanizomenoides*, *Cuspidothrix issatschenkoi*, and *Raphidiopsis raciborskii* [9,10,11,12], were recorded in Lake Sztynorckie at all sampling sites, especially in the hottest summer month. Previously, they were known as only tropical cyanobacteria, but during the last decades, their presence expanded to temperate stagnant freshwaters [57]. This phenomenon was also noted in other aquatic ecosystems [13]. These species grew in both high and low light conditions and were accompanied by low to high phytoplankton biomass. All three species were not recorded as the main bloom contributors but only as accompanying bloom-forming species. However, they are predicted to become the main nuisance species of the future due to global warming and worldwide eutrophication [10,57]. The present study indicated that the recreational pressure, along with a temperature rise, could affect the enhanced cyanobacterial blooms in small water bodies. It concerns nostocalean species in general, and especially the invasive species due to a more northward expansion.

It is well known that *C. issatschenkoi* is primarily an anatoxin-a-producing cyanobacteria [11,18,19], but it is capable of producing other cyanotoxins. *R. raciborskii*, in turn, is responsible for cylindrospermopsin (CYN) production [21]. *C. issatschenkoi* and *R. raciborskii* are also commonly known as producers of saxitoxins (STXs) [58]. CYN can also be produced by *S. aphanizomenoides*, since it possesses the appropriate gene [22,59]. This toxin’s effects can manifest with a delay in the animal and human body due to an inhibition of protein synthesis or even genotoxicity [60].

The invasive cyanobacteria, including the toxin-producing and possibly bloom-forming species, can also affect the local biodiversity across Europe [12]. In Lake Sztynorckie, the loss of planktonic biodiversity was manifested by relatively low species richness of phytoplankton (up to 76 species) and zooplankton (37 species) and the changes in their structure towards more eutrophic conditions of the lake. Situations such as this, with additional blooming events due to toxin-producing cyanobacterial species throughout the growing season, can affect the water quality of other lakes connected with Lake Sztynorckie. It was confirmed that the lake conditions create a possible niche for the development of invasive species, and furthermore, their migration with ships, boats, or birds can occur simultaneously.

Previous studies on cyanobacteria in the lakes within the Great Masurian Lakes District carried out at the turn of the 20th and 21st century [2,51,61] did not show any presence of *S. aphanizomenoides* or *R. raciborskii* in phytoplankton. Lake Sztynorckie has a direct connection to Lake Dargin, which was assessed as one of the cleanest lakes within the Great Masurian Lakes System. Current studies indicated that these invasive and potentially toxic species occur in Lake Sztynorckie. Additionally, this lake with a marina zone is linked directly via a navigable trail to other recreationally attractive Masurian lakes as well. In the summer of 2020, relatively high tourism pressure was recorded, expressed as the number of users in the port of Lake Sztynorckie. These tourists sailed to other Masurian lakes and could potentially affect the spread of harmful cyanobacteria over all these lakes.

The overall tendency in phytoplankton density and biomass in Lake Sztynorckie was similar on a temporal scale, with the minimum in October and the maximum in August, and showed a similar domination structure. In contrast, such temporal changes were different for zooplankton, whose maximum biomass due to the high share of Cladocera and Copepoda in June coincided with the relatively low phytoplankton biomass. As a notable exception, these relations were different in the inflow zone (site S2). The maximum total zooplankton biomass, and domination of Rotifera both in biomass and density, were found at site S2 in August, whereas Cyanobacteria-dominated phytoplankton biomass was non-typically lower in August than in April compared to other sampling sites.

## 5. Conclusions

In recreationally used Lake Sztynorckie, phytoplankton formed biomass in the range of 28–70 mg L^−1^, which is typical of strong blooms which lasted throughout the whole growing season. It is characteristic of lakes with the increased trophy. The dominant filamentous cyanobacteria *Pseudanabaena limnetica*, *Limnothrix redekei*, *Planktolyngbya limnetica*, and *Planktothrix agarhii* formed blooms in the whole water column. Among them, three invasive nostocalean species, *Sphaerospermopsis aphanizomenoides*, *Cuspidothrix issatschenkoi*, and *Raphidiopsis raciborskii*, were also frequently found in phytoplankton. They can pose a serious new threat not only to the ecosystem—but also to humans. All these cyanobacteria species are known to be toxic or potentially toxic due to their production of cyanotoxins, such as microcystins, saxitoxins, anatoxin-a, and cylindrospermopsins, having hepatotoxic, cytotoxic, neurotoxic, and dermatoxic effects. Based on phytoplankton, water quality assessment indicated a bad ecological status, while that based on zooplankton confirmed highly meso-eutrophic or mesotrophic levels of Lake Sztynorckie. The phyto- and zooplankton biomass relations, in turn, were characteristic of very low trophic efficiency. The dominant species in both groups of planktonic organisms were typical of highly or less eutrophic conditions.

## Figures and Tables

**Figure 1 ijerph-20-05063-f001:**
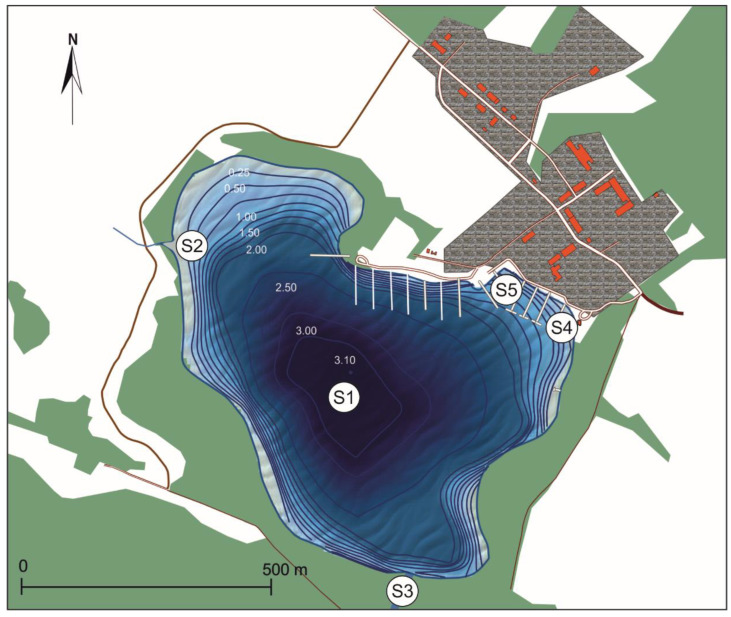
Bathymetric map and location of sampling sites in Lake Sztynorckie (after IRS 1950, modified); site S1–profundal zone; site S2—inflow zone; site S3—channel zone; site S4—fuel zone; site S5—mooring zone.

**Figure 2 ijerph-20-05063-f002:**
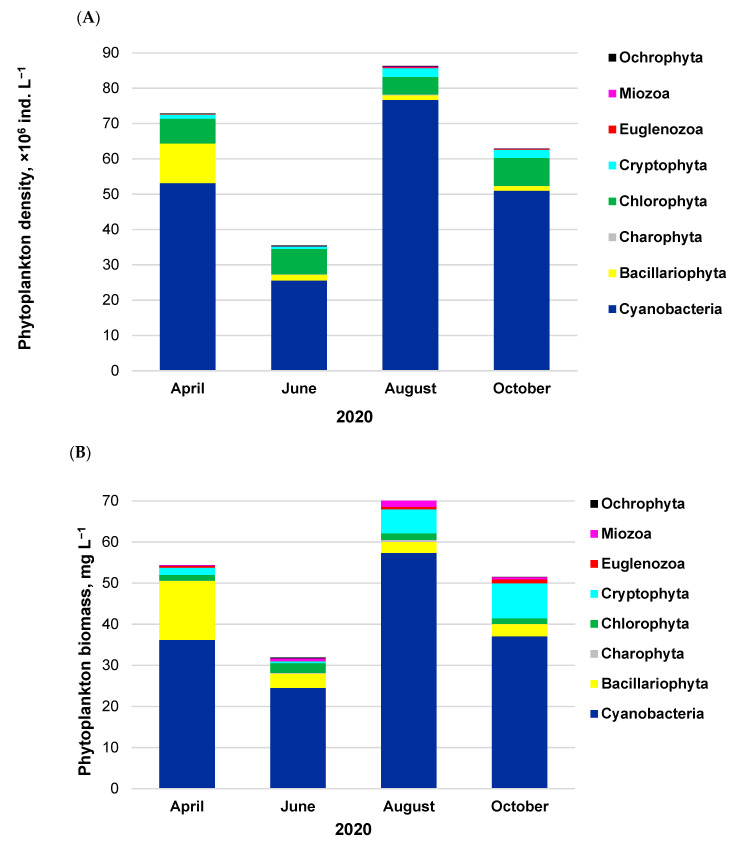
Phytoplankton density (**A**) and biomass (**B**) in Lake Sztynorckie (representative site in profundal zone S1) with an emphasis on each taxonomic group.

**Figure 3 ijerph-20-05063-f003:**
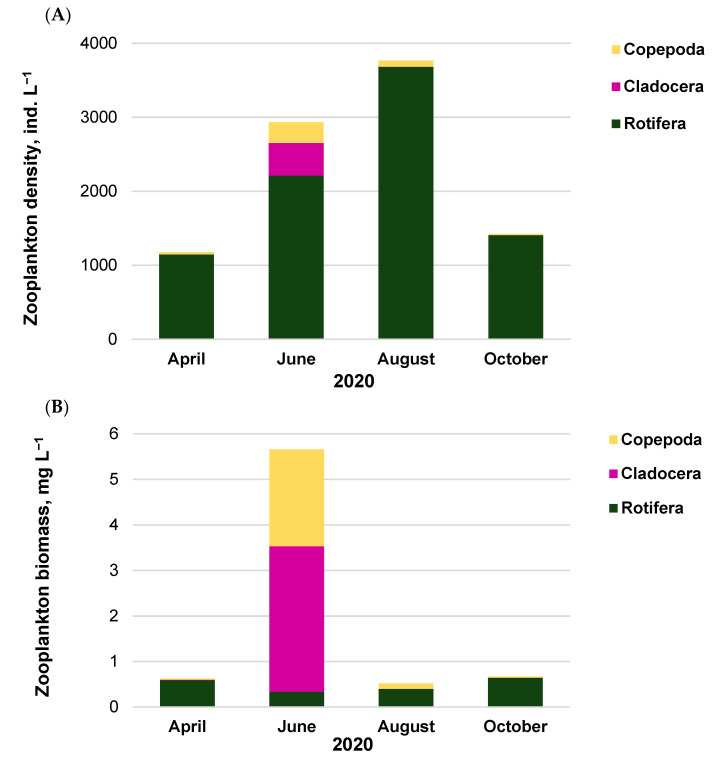
Zooplankton density (**A**) and biomass (**B**) in Lake Sztynorckie (representative site in profundal zone, S1) with an emphasis on each taxonomic group.

**Figure 4 ijerph-20-05063-f004:**
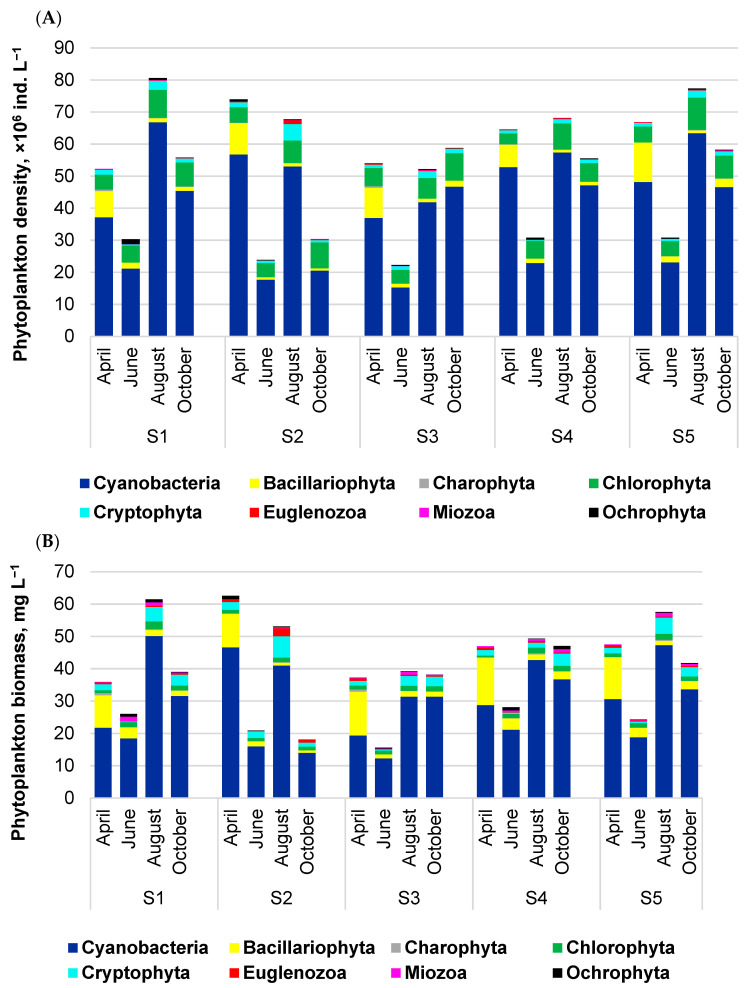
Spatial distribution of phytoplankton density (**A**) and biomass (**B**) in surface waters of Lake Sztynorckie. Sampling sites: S1—profundal zone; S2—inflow zone; S3—channel zone; S4—fuel zone; S5—mooring zone.

**Figure 5 ijerph-20-05063-f005:**
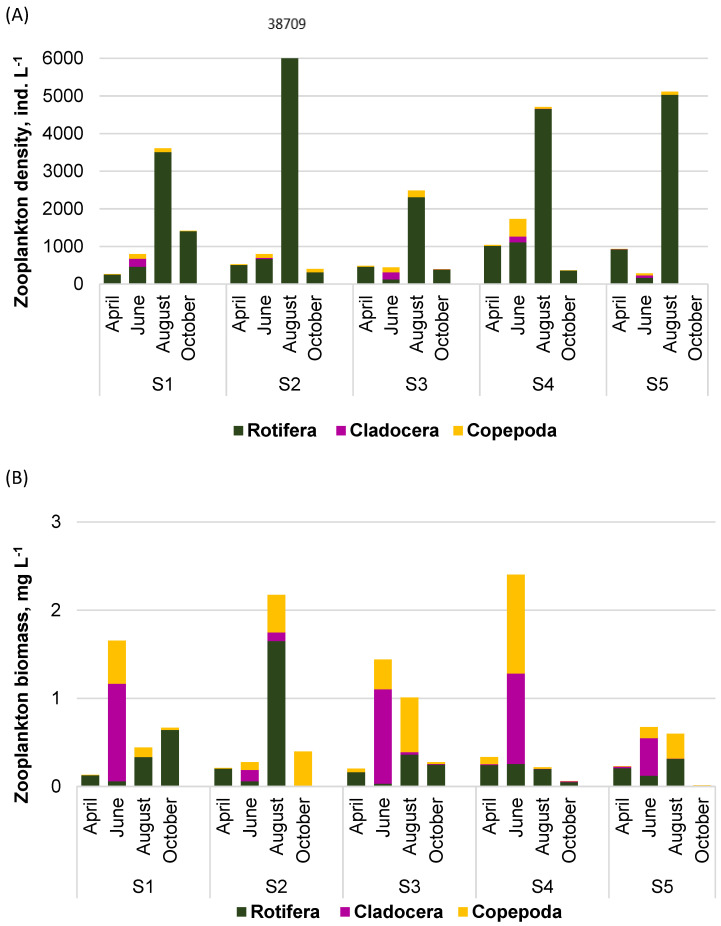
Spatial distribution of zooplankton density (**A**) and biomass (**B**) in surface waters of Lake Sztynorckie. Sampling sites: S1—profundal zone; S2—inflow zone; S3—channel zone; S4—fuel zone; S5—mooring zone.

**Figure 6 ijerph-20-05063-f006:**
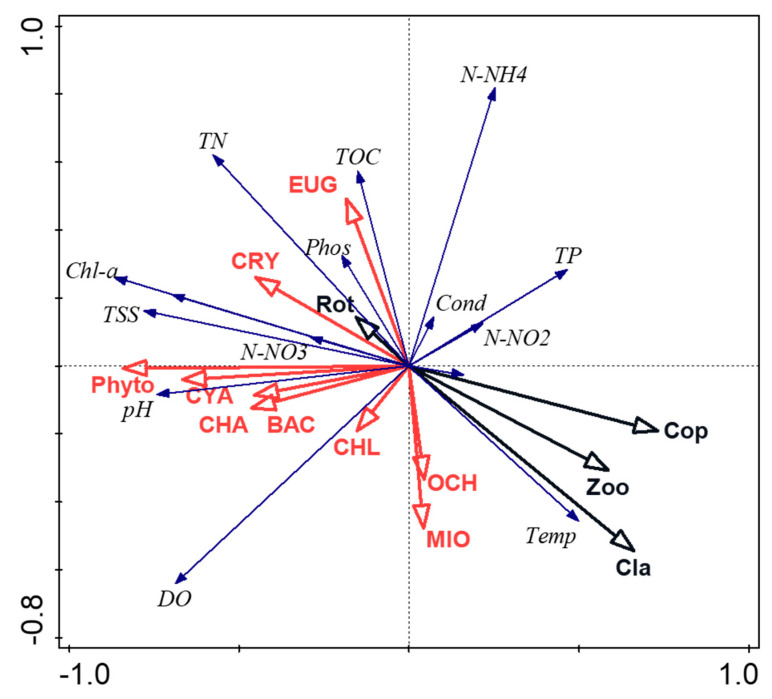
The redundancy analysis (RDA) relative to phytoplankton, zooplankton, and environmental parameters. Codes: Phyto—phytoplankton; CYA—Cyanobacteria; BAC—Bacillariophyta; CHA—Charophyta; CHL—Chlorophyta; CRY—Cryptophyta; EUG—Euglenozoa; MIO—Miozoa; OCH—Ochrophyta; Zoo—Zooplankton; Cla—Cladocera; Cop—Copepoda; Rot—Rotifera; Temp—water temperature; DO—dissolved oxygen; Cond—conductivity; pH, N-NO2—nitrites; N-NO3—nitrates; N-NH4—ammonium nitrogen; TN—total nitrogen; TP—total phosphorus; Phos—phosphates; TOC—total organic carbon; Chl-a—chlorophyll a; and TSS—total suspended solids. Red arrows indicate total phytoplankton and its groups, black arrows indicate total zooplankton and its groups, blue arrows indicate physicochemical parameters.

**Figure 7 ijerph-20-05063-f007:**
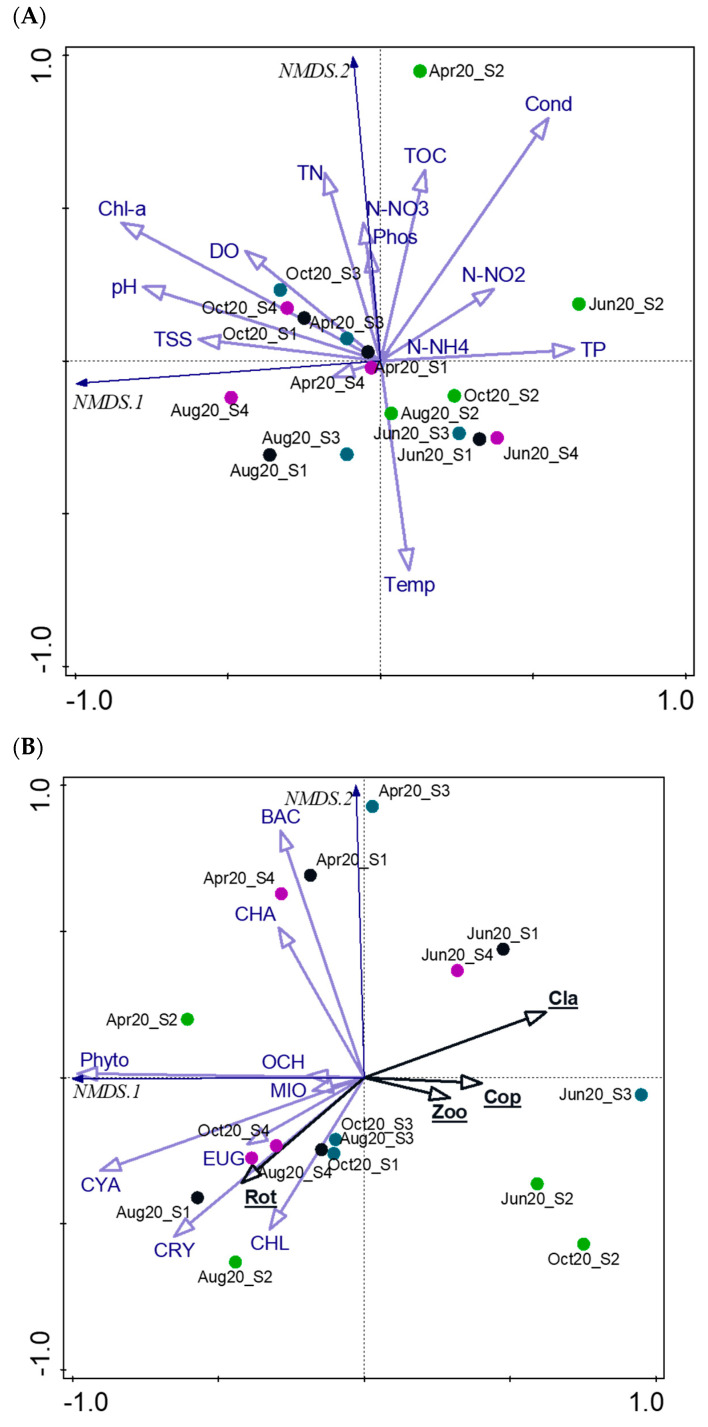
The non-metric multidimensional scaling (NMDS) plots based on environmental parameters (**A**) and phytoplankton and zooplankton (**B**). Codes: Temp—water temperature; DO—dissolved oxygen; Cond—conductivity; pH, N-NO2—nitrites; N-NO3—nitrates; N-NH4—ammonium nitrogen; TN—total nitrogen; TP—total phosphorus; Phos—phosphates; TOC—total organic carbon; Chl-a—chlorophyll a; and TSS—total suspended solids; Phyto—phytoplankton; CYA—Cyanobacteria; BAC—Bacillariophyta; CHA—Charophyta; CHL—Chlorophyta; CRY—Cryptophyta; EUG—Euglenozoa; MIO—Miozoa; OCH—Ochrophyta; Zoo—Zooplankton; Cla—Cladocera; Cop—Copepoda; Rot—Rotifera; Apr—April; Jun—June; Aug—August; Oct—October; S1—profundal zone; S2—inflow zone; S3—channel zone; S4—fuel zone. Blue arrows indicate (**A**) physicochemical parameters and (**B**) total phytoplankton and its groups, black arrows indicate total zooplankton and its groups; black circle – samples from site S1, green circle—samples from site S2, blue circle—samples from site S3, purple circle—samples from site S4.

**Table 1 ijerph-20-05063-t001:** Dominant species (≥5% of total biomass) in phytoplankton of Lake Sztynorckie during the whole growth season of 2020.

Date	Species	Functional Classification
April	*Limnothrix redekei* (30%)	S1
	*Pseudanabaena limnetica* (24%)	S1
	*Ulnaria acus* (20%)	D
	*Planktolyngbya limnetica* (12%)	S1
	*Ulnaria ulna* (5%)	D
June	*Limnothrix redekei* (48%)	S1
	*Pseudanabaena limnetica* (10%)	S1
	*Aphanizomenon gracile* (8%)	H1
	*Planktolyngbya limnetica* (5%)	S1
August	*Pseudanabaena limnetica* (27%)	S1
	*Aphanizomenon gracile* (23%)	H1
	*Planktolyngbya limnetica* (13%)	S1
	*Raphidiopsis raciborskii* (8%)	SN
	*Limnothrix redekei* (7%)	S1
October	*Pseudanabaena limnetica* (28%)	S1
	*Cryptomonas erosa* (16%)	Y
	*Raphidiopsis raciborskii* (15%)	SN
	*Planktolyngbya limnetica* (12%)	S1
	*Limnothrix redekei* (11%)	S1
	*Ulnaria acus* (5%)	D

Habitat template [45,46]: S1—turbid mixed environments. This codon includes only shade-adapted cyanobacteria; D—shallow turbid waters including rivers; H1—eutrophic, both stratified and shallow lakes with low nitrogen content; S_N_—warm mixed environments; Y—this codon, mostly including large cryptomonads but also small dinoflagellates, refers to a wide range of habitats which reflect the ability of its representative species to live in almost all lentic ecosystems with low grazing pressure.

**Table 2 ijerph-20-05063-t002:** Dominant species (≥5% of total biomass) in zooplankton of Lake Sztynorckie during the whole growing season of 2020.

Date	Species	Functional Classification
April	*Polyarthra longiremis* (47%)	Eutrophy
	*Keratella quadrata* (12%)	Eutrophy
	*Asplanchna priodonta* (9%)	Mesotrophy
	*Paracyclops* sp. (6%)	-
June	*Bosmina longispina* (31%)	Mesotrophy
	*Bosmina longirostris* (22%)	Eutrophy
	*Macrocylops albidus* (15%)	Mesotrophy
	*Thermocyclops* sp. (9%)	Eutrophy
August	*Polyarthra longiremis* (40%)	Eutrophy
	*Anuraeopsis fisa* (22%)	Eutrophy
	*Macrocylops albidus* (21%)	Mesotrophy
	*Cyclopoida nauplii* (13%)	Mesotrophy
	*Trichocerca capucina* (6%)	Mesotrophy
	*Trichocerca pusilla* (6%)	Eutrophy
October	*Asplanchna priodonta* (36%)	Mesotrophy
	*Polyarthra euryptera* (23%)	Mesotrophy
	*Mesocyclops leuckartii* (16%)	Mesotrophy
	*Eudiaptomus gracilis* (8%)	Eutrophy

**Table 3 ijerph-20-05063-t003:** Appearance of invasive cyanobacteria species (≥5% of the total biomass) in phytoplankton of Lake Sztynorckie during the whole growing season in 2020.

Date	Sampling Site	*Sphaerospermopsis aphanizomenoides*	*Cuspidothrix issatschenkoi*	*Raphidiopsis raciborskii*
April	S1	−	−	−
	S2	−	−	−
	S3	−	−	−
	S4	−	+	−
	S5	−	+	−
June	S1	−	−	−
	S2	−	−	−
	S3	−	−	−
	S4	+	−	−
	S5	+	−	−
August	S1	+	+	+
	S2	+	+	+
	S3	+	+	+
	S4	+	+	+
	S5	+	+	+
October	S1	−	+	−
	S2	−	+	−
	S3	+	+	−
	S4	−	+	+
	S5	−	+	−

Codes: S1—profundal zone; S2—inflow zone; S3—channel zone; S4—fuel zone; and S5—mooring zone; + present, − absent.

## Data Availability

The data presented in this study are available in this article and can be provided on request.

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
