# Peer review of "Ecological Implications in a Human-Impacted Lake—A Case Study of Cyanobacterial Blooms in a Recreationally Used Water Body"

_ijerph, 2023, doi:10.3390/ijerph20065063_

Round 1
Reviewer 1 Report
In the title you imply you are measuring recreation and water pollution, yet there is no data on recreations relationship to pollution,other than the number of people using the port.
In line 3 of the introduction and line 71 of the text you state 'ensure that recreational pressure may enhance cyanobacterial blooms' There is no statistical comparison or correlation between recreation and the phytoplankton data. The phrase you use does not make sense.
The ecological information is clear, but there is a reference to microcystins, which you do not report as having been measured.
Author Response
The authors of the manuscript thank the Reviewer for his valuable comments. The work has been very carefully corrected and refined as suggested. The authors hope that the current version meets the requirements for article in International Journal of Environmental Research and Public Health.
- In the title you imply you are measuring recreation and water pollution, yet there is no data on recreations relationship to pollution, other than the number of people using the port.
Response
Thank you for your comment. We changed the title into: “Ecological implications in a human impacted lake – a case study of cyanobacterial blooms in recreationally used water body”.
- In line 3 of the introduction and line 71 of the text you state 'ensure that recreational pressure may enhance cyanobacterial blooms' There is no statistical comparison or correlation between recreation and the phytoplankton data. The phrase you use does not make sense.
Response
Thank you for this comment. We changed the second aim into: The second aim was to analyze the aspect of recreational pressure which may enhance cyanobacterial blooms, and as a consequence the negative changes and loss of planktonic biodiversity.”
Thus, in the improved version the aims were included as follows:
“Therefore, the aims of this study were to (1) describe the phytoplankton and zoo-plankton assemblages with a special attention on invasive toxin-producing cyanobacteria species, and (2) investigate if cyanobacterial blooms linked with loss of planktonic biodiversity and changes in the whole phytoplankton and zooplankton structures may be enhanced by recreational pressure.”
- The ecological information is clear, but there is a reference to microcystins, which you do not report as having been measured.
Response
Thank you for this comment. We gave the reference to microcystins, because the literature date confirmed that one of a special attention cyanobacterial species Sphaerospermopsis aphanizomenoides can produce microcystins. Therefore, we cannot exclude this aspect in our study. In article we gave the information on microcystins production by this species noted in in Moroccan lake as well as in Polish lakes as examples:
“The findings of Sabour et al. [22] and Dziga et al. [23] confirm the ability of S. aphani-zomenoides to produce three variants of microcystins.”
Reviewer 2 Report
This is a study limited to a small water body. Some details should be improved before acceptance.
Line 67-the objectives should be further elaborated. How recreational activities can impact waterbodies? Examples? References? The specific lake to be studied should also be presented briefly here, and not just in section 2.
Line 71 -the word «ensure» is misplaced here. It seems the authors would have caused pollution on purpose to ‘ensure’ cianoHABs would grow. Maybe ‘research if’ is more appropriate.
Line 78-is this the lake catchment named IZ on Fig. 1?
Line 80-if there is only one canal, leading to lake Dargin, how can water flow to lake Sztynorckie?
Line 86-93-the locations were named twice, by letters and by numbers. Please choose only one format for simplicity.
Line 102/110-«standard methods» is rather vague for the international community. Please detail.
Line 127-what does the word «term» means? Is this a ‘thermal bottle’?
Line 161-section 3.1- In this section, the phyto- and zoo- assemblages were described. The research had 5 different sampling locations. What location was described in this section 3.1? In figures 2 and 3, zone S1 was referred to.
Line 251-256-some of the concentrations described here are not visualized in Fig. 5a. The y-axis scale ends at 6000 while the value reported for station S2 was 38709 ind/L.
Author Response
REVIEWER 2
The authors of the manuscript thank the Reviewer for his valuable comments. The work has been very carefully corrected and refined as suggested. The authors hope that the current version meets the requirements for article in International Journal of Environmental Research and Public Health.
This is a study limited to a small water body. Some details should be improved before acceptance.
- Line 67-the objectives should be further elaborated. How recreational activities can impact waterbodies? Examples? References? The specific lake to be studied should also be presented briefly here, and not just in section 2.
Response:
Thank you for these comments and suggestions. We supplied improved manuscript with the references about the impact of recreational activities on water quality.
“The negative effects of recreational activities on water quality was confirmed in few findings [26,27]. The small, shallow and recreationally used Lake Sztynorckie was selected to present the pattern of impact on the water environment. In 2020, the tourism pressure, the number of users (chartering and passengers) from the port was as follows: in April – 583 people, in May – 5,015 people, June – 14,724 people, in July – 15,880 people, and in August – 15,822 people (data from the port Owner's register). Tourist activities, including chartering the yachts with the function of transport and accommodation for passengers, occur directly to enhance the pollutants in the water [28].”
- Baoying N.; Yuanqing H. Tourism Development and Water Pollution: Case Study in Lijiang Ancient Town, China Population. Resources and Environment, 2007, 17(5), 123-127, https://doi.org/10.1016/S1872-583X(08)60006-6.
- Lenart-Boroń A.M.; Boroń P.M.; Prajsnar J.A.; Guzik M.W.; Żelazny M.S.; Pufelska M.D.; Chmiel, M.J. COVID-19 lockdown shows how much natural mountain regions are affected by heavy tourism. Sci. Tot. Environ. 2022, 806 (3), 151355. https://doi.org/10.1016/j.scitotenv.2021.151355.
- Kiersztyn, B.; Chróst, R.; KaliÅ„ski, T.; Siuda, W.; Bukowska, A.; Kowalczyk, G.; Grabowska, K.; Structural and functional microbial diversity along a eutrophication gradient of interconnected lakes undergoing anthropopressure. Sci. Rep. 2019, 9, 11144. https://doi.org/10.1038/s41598-019-47577-8
- Line 71 -the word «ensure» is misplaced here. It seems the authors would have caused pollution on purpose to ‘ensure’ cianoHABs would grow. Maybe ‘research if’ is more appropriate.
Response:
Thank you for this comment. We changed the second aim of study according to suggestions.
“Therefore, the aims of this study were to (1) describe the phytoplankton and zoo-plankton assemblages with a special attention on invasive toxin-producing cyanobacteria species, and (2) investigate if cyanobacterial blooms linked with loss of planktonic biodiversity and changes in the whole phytoplankton and zooplankton structures may be enhanced by recreational pressure.”
- Line 78-is this the lake catchment named IZ on Fig. 1?
Response:
Thank you for this comment. The IZ is the inflow zone on Fig. 1 – it is a permanent inflow to the Lake Sztynorckie, i.e. a drainage ditch which supplies water from the catchment area. In improved version we unified the sampling sites as S1, S2, S3, S4 and S5. Now, S2 means the sampling sites within IZ.
- Line 80-if there is only one canal, leading to lake Dargin, how can water flow to lake Sztynorckie?
Response:
Thank you for this comment. We clarified the information and detailed the water flow: inflow to the lake and outflow from Lake Sztynorckie, as follows:
“According to the Computer Map of the Hydrographic Division of Poland (The State Water Holding Polish Waters), Lake Sztynorckie has 1 main tributary which is divided into three sections: (1) a permanent inflow (a drainage ditch from the catchment area), (2) flow through Lake Sztynorckie, and (3) a channel connecting Lake Sztynorckie with Lake Dargin.”
- Line 86-93-the locations were named twice, by letters and by numbers. Please choose only one format for simplicity.
Response:
Thank you for this comment. In improved version we unified the codes and we used only codes of sampling sites: S1, S2, S3 S4 and S5 throughout the whole body text as locations names.
- Line 102/110-«standard methods» is rather vague for the international community. Please detail.
Response:
Thank you for this comment. We clarified “standard methods” as follows:
“The integrated water samples were taken at the deepest site S1 from the whole water column (at one-meter intervals, mixed in the bucket, and then the final water sample was taken).”
- Line 127-what does the word «term» means? Is this a ‘thermal bottle’?
Response
Thank you for this comment. The word “terms” means “dates” or “sampling occasions”. In improved version we decided to use “dates”.
- Line 161-section 3.1- In this section, the phyto- and zoo- assemblages were described. The research had 5 different sampling locations. What location was described in this section 3.1? In figures 2 and 3, zone S1 was referred to.
Response
Thank you for this comment. The section 3.1. concerns the phytoplankton and zooplankton assemblages within routine monitoring system. It means, that it was selected the representative site for the lake, and it was site in profundal zone marked as S1. For checking the spatial variations, we selected 5 different sampling locations.
We clarified this in section 2. Materials and Methods as follows:
“The water samples for standard analysis used in routine monitoring system of water quality were taken from a representative site – S1 in the deepest point of the lake. Additionally, the sub-surface water samples were taken from five sites: 1-5, i.e. site S1 – profundal zone as the deepest point of the lake, site S2 – inflow zone near channel between the pumping station and Lake Sztynorckie (outflow from the retention reservoir), site S3 – channel zone including Sztynorcki channel which connects Lake Sztynorckie with Lake Dargin, and sites associated with harbour i.e. site S4 – fuel zone and site S5 – mooring zone. These five sampling sites were selected to check the impact of human activities on waters.”
- Line 251-256-some of the concentrations described here are not visualized in Fig. 5a. The y-axis scale ends at 6000 while the value reported for station S2 was 38709 ind/L.
Response:
Thank you for this comment. We added the value of 38709 ind/L to the graph without the changes of y-axis scale because we want to keep other data visible. Such high value was the exception in comparison to other values of zooplankton density and included: Rotifera – 38504 ind/L, Cladocera – 34 ind/L and Copepoda – 171 ind/L.
Reviewer 3 Report
The authors have presented a research on Lake Sztynorckie (Poland), primarily about phytoplankton and zooplankton community changes during 2020 in several locations of the lake. The name of the manuscript could be adjusted to depict more precisely the topic. It is very vague and implies a lot of results, although it is mostly focused on a few aspects of the lake parameters during one year.
First of all, English in the introduction part is a bit problematic and it is hard to follow the manuscript. As I am not an English native speaker, I suggest that manuscript should be checked in more detail with a native speaker. It must be a priority before sending back the corrected manuscript.
Overall, manuscript offers some new data and insight into the status of the lake, but it is quite repetitive and not very innovative. Serious review of the English should be done before publication. More specific corrections are provided within the manuscript. Finally, although I have some reservation regarding the topic of the research, information provided could be informative so I’d recommend publication after corrections.

Author Response
The authors of the manuscript thank the Reviewer for his valuable comments. The work has been very carefully corrected and refined as suggested. The authors hope that the current version meets the requirements for article in International Journal of Environmental Research and Public Health.
- The authors have presented a research on Lake Sztynorckie (Poland), primarily about phytoplankton and zooplankton community changes during 2020 in several locations of the lake. The name of the manuscript could be adjusted to depict more precisely the topic. It is very vague and implies a lot of results, although it is mostly focused on a few aspects of the lake parameters during one year.
Response
Thank you for this comment. We changed the title into: “Ecological implications in a human impacted lake – a case study of cyanobacterial blooms in recreationally used water body”.
- First of all, English in the introduction part is a bit problematic and it is hard to follow the manuscript. As I am not an English native speaker, I suggest that manuscript should be checked in more detail with a native speaker. It must be a priority before sending back the corrected manuscript.
Response
Thank you for this comment. The manuscript was checked in more detail with a native speaker.
- Overall, manuscript offers some new data and insight into the status of the lake, but it is quite repetitive and not very innovative. Serious review of the English should be done before publication. More specific corrections are provided within the manuscript. Finally, although I have some reservation regarding the topic of the research, information provided could be informative so I’d recommend publication after corrections.
Response
Thank you for the comments and recommendation. We improved the whole article according to specific corrections which were provided in within the manuscript. The English language was checked by the professional.
Round 2
Reviewer 1 Report
Revisions cover the issues raised.
Author Response
Dear Reviewer
Thank you very much for your comment.
All sugestions were included in the revised version of our manuscritpt.
Reviewer 2 Report
The authors have improved the manuscript. However, it still requires attention as outlined below:
Line 92- novel reference ‘Chen’ was not placed in the reference list.
Line 97- «ifcyanobacterial», the new text is missing the adequate spacing. Please check throughout the text.
Line 135-the use of «standard methods» was not improved. Are these ‘internal’ methods used by their Institution? Others? A known international method?
The figures in the pdf were compressed and have very low resolution.
Author Response
Dear Reviewer
Response to Reviewer 1 Comments
Thank you vey much for your comments.
The English languages was improved. The detailed responses were given below.
Point 1: Line 92- novel reference ‘Chen’ was not placed in the reference list.
Response 1: Thank you for this comment. We added teo references of Chen et al. and Dokulil which were missing in reference list:
- Chen, M.J.; Balomenou, C.; Nijkamp, P.; Poulaki, P.; Lagos, D. The Sustainability of Yachting Tourism: A Case Study on Greece. International Journal of Research in Tourism and Hospitality 2016, 2(2), 42-49. http://dx.doi.org/10.20431/2455-0043.0202005
- Dokulil, M.T. Environmental Impacts of Tourism on Lakes. In Eutrophication: Causes, Consequences and Control. Ansari, A.A.; Gill, S.S., Eds. Springer Science+Business Media Dordrecht, 2014, 81-88. DOI 10.1007/978-94-007-7814-6_7
Point 2: Line 97- «ifcyanobacterial», the new text is missing the adequate spacing. Please check throughout the text.
Response 2: Thank you for this comment. We added the missing space.
Point 3: Line 135-the use of «standard methods» was not improved. Are these ‘internal’ methods used by their Institution? Others? A known international method?
Response 3: Thank you for this comment. We claryfied that standard methods using for routine monitoring of biological elements icluded methods for phytoplankton as follows: “These methods include the integrated water samples which were taken at the deepest site S1 from the whole water column, i.e. at one-meter intervals, mixed in the bucket, and then the final water sample was taken. Additional samples were taken from the subsurface (0.5 m) water layers. The phytoplankton samples for biomass analysis were not concentrated where-as zooplankton samples were concentrated (25 L) using a 55 µm mesh size plankton net. These samples were fixed with Lugol’s solution and 96% ethanol. Furthermore, for phytoplankton taxonomic analysis, the samples were taken using a 10 µm mesh size plankton net and they were analyzed live after sampling.”
Point 4: The figures in the pdf were compressed and have very low resolution.
Response 4: Thank you for this comment. The resolution of figures was improved and they were sent in additional Word file with figures.